# REASONING-ENHANCED OBJECT-CENTRIC LEARNING FOR VIDEOS

## ABSTRACT

Object-centric learning aims to break down complex visual scenes into more manageable object representations, enhancing the understanding and reasoning abilities of machine learning systems toward the physical world. Recently, slot-based video models have demonstrated remarkable proficiency in segmenting and tracking objects. Although most modules in these models are well-designed, they overlook the importance of the effective reasoning module. In the real world, especially in complex scenes, reasoning and predictive abilities play a crucial role in human perception and object tracking; in particular, these abilities are closely related to human intuitive physics. Inspired by this, we designed a novel reasoning module called the Slot-based Time-Space Transformer with Memory buffer (STATM) to enhance the model's perception ability in complex scenes. The memory buffer primarily serves as storage for slot information from upstream modules, akin to human memory or field of view. The Slot-based Time-Space Transformer makes predictions through slot-based spatiotemporal attention computations and fusion. We demonstrated that the improved deep learning model exhibits certain degree of rationality imitating human behavior. This has crucial implications for understanding the relationship between deep learning and human cognition, especially in the context of intuitive physics.

## 1 INTRODUCTION

Objects are the fundamental elements that constitute our world, which adhere to the fundamental laws of physics. Humans learn through activities such as observing the world and interacting with it. They utilize the knowledge acquired via these processes for reasoning and prediction. All these aspects are crucial components of human intuitive physics (Lake et al., 2017; Kubricht et al., 2017; Riochet et al., 2018; Smith, 2019). Therefore, object-centric research is pivotal for comprehending human cognitive processes and for developing more intelligent artificial intelligence (AI) systems. By studying the properties, movements, interactions, and behaviors of objects, we can uncover the ways and patterns in which humans think and make decisions in the domains of perception, learning, decision-making, and planning. This contributes to the advancement of more sophisticated machine learning algorithms and AI systems, enabling them to better understand and emulate human intuitive physical abilities (Janner et al., 2019; Tang et al., 2023).

Recently, the representative SAVi (Kipf et al., 2021) and SAVi++ (Elsayed et al., 2022) models have demonstrated impressive performance in object perception. SAVi (Slot Attention for Video) employed optical flow as a prediction target and leveraged a small set of abstract hints as conditional inputs in the first frame to acquire object-centric representations of dynamic scenes. SAVi++ (Towards End-to-End Object-Centric Learning from Real-World Videos) enhanced the SAVi by integrating depth prediction and implementing optimal strategies for architectural design and data augmentation. Both SAVi and SAVi++ execute two steps on observed video frames: a prediction step and a correction step. The correction step uses inputs to update the slots. The prediction step uses the slots information of the objects provided by the correction step for prediction. The predictor's output initializes the correction process in the subsequent time step, ensuring the model's consistent ability to track objects over time.

The two main steps of such a model operate in a positive feedback loop. The more accurate the predictions, the better the corrections become. Consequently, the more accurate the corrections, the

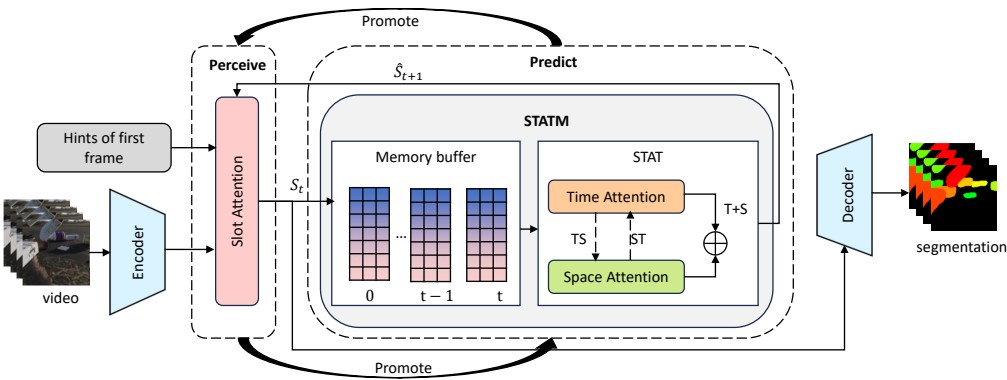

Figure 1: Slot-based Time-Space Transformer with Memory buffer architecture overview.

more precise the information provided for the prediction step is, leading to better predictions. Therefore, having a reasonable and efficient predictor is crucial for the entire model. In real-world scenarios, humans also engage in prediction as a crucial aspect of their object perception and tracking, but human prediction behaviors often involve more intricate processes. Humans typically combine the motion state of an object with the interactions of other objects to predict possible future states and positions of the object. The object's motion state is inferred by humans using their common sense from the object's past positions over a period of time. In so doing, humans enhance their ability to recognize and track relevant objects within complex scenes, which is an integral component of human intuitive physics (Sudderth, 2006; Ullman et al., 2017; Mitko & Fischer, 2020). In simpler environments, considering our ability to instantly recognize objects in a single shot, the potential of humans in this regard may be underestimated. The prediction step in SAVi and SAVi++ is similar to human inference, but the predictor module in SAVi and SAVi++ is somewhat simplistic, as it relies solely on single-frame information from the current time step for prediction.

Drawing inspiration from human behavior, we introduce a novel prediction module aimed at enhancing slot-based models for video. This module comprises two key components: 1) Slot-based Memory Buffer: primarily designed to store historical slot information obtained from the upstream modules; and 2) Slot-based Time-Space Transformer Module: designed by applying spatiotemporal attention mechanisms to slots for inferring the temporal motion states of objects and calculating spatial objects interactions, which also integrates time and space attention results. We term the proposed model as *Slot-based Time-Space Transformer with Memory buffer* (STATM). Upon substituting the prediction module of SAVi and SAVi++ into the STATM, we observe distinct impacts of different spatiotemporal fusion methods on SAVi and SAVi++. By employing an appropriate fusion method and memory buffer sizes, we observed a significant enhancement in the object segmentation and tracking capabilities of SAVi and SAVi++ on videos containing complex backgrounds and a large number of objects per scene.

## 2 RELATED WORK

**Object-centric Learning.** In recent years, object-centric learning has emerged as a significant research direction in computer vision and machine learning. It aims to enable machines to perceive and understand the environment from an object-centered perspective, thereby constructing more intelligent visual systems. There is a rich literature on this research, including SQAIR (Kosiorek et al., 2018), R-SQAIR (Stanić & Schmidhuber, 2019), SCALOR (Jiang et al., 2019), Monet (2019), OP3 (Veerapaneni et al., 2020), ViMON (Weis et al., 2020), PSGNet (Bear et al., 2020), SIMONe (Kabra et al., 2021), and others (Kahneman et al., 1992; Kipf et al., 2019; Zhang et al., 2022; Xie et al., 2022; Seitzer et al., 2023; Zadaianchuk et al., 2023; ZHANG et al., 2023; Nakano et al., 2023; Jia et al., 2023). Slot-based Models represent a prominent approach within object-centric learning. They achieve this by representing each object in a scene as an individual slot, which is used to store object features and attributes (Locatello et al., 2020; Kumar et al., 2020; Zoran et al., 2021; Singh et al., 2021; Yang et al., 2021; Zoran et al., 2021; Ye et al., 2021; Hassanin et al., 2022; Wang et al., 2023; Heravi et al., 2023; Wu et al., 2023).

**Prediction and Inference on Physics.** The implementation of object-centric physical reasoning is crucial for human intelligence and is also a key objective in artificial intelligence. Interaction Network (Battaglia et al., 2016) as the first general-purpose learnable physics engine, is capable of performing reasoning tasks centered around objects or relationships. Another similar study is the Neural Physics Engine (Chang et al., 2016). On the other hand, Visual Interaction Networks (Watters et al., 2017) can learn physical laws from videos to predict the future states of objects. Additionally, there are many models developed based on this foundation (Engelcke et al., 2019; Henderson & Lampert, 2020; Chen et al., 2021; Dittadi et al., 2021; Jusup et al., 2022; Meng et al., 2022; Piloto et al., 2022; Singh et al., 2022; Driess et al., 2023; Cornelio et al., 2023). In order to achieve a deeper understanding of commonsense intuitive physics within artificial intelligence systems, Piloto et al. (2022) have built a system capable of learning various physical concepts, albeit requiring access to privileged information such as segmentation. Our research primarily aims to construct an object-centric system for object perception, learning of physics, and reasoning.

**Slot-based Attention and spatiotemporal Attention.** Our current work is closely related to slot-based attention and spatiotemporal attention. There are a lot of works related to slot-based attention (Locatello et al., 2020; Hu et al., 2020; Kumar et al., 2020; Zoran et al., 2021; Singh et al., 2021; Yang et al., 2021; Zoran et al., 2021; Ye et al., 2021; Hassanin et al., 2022; Wang et al., 2023; Heravi et al., 2023; Wu et al., 2023). Spatiotemporal attention mechanisms are particularly effective in handling video data or time-series data, allowing networks to understand and leverage relationships between different time steps or spatial positions (Li et al., 2020; Luo et al., 2021). Currently, they find wide applications in various fields such as video object detection and tracking (Lin et al., 2021; Chen et al., 2022), action recognition (Yang et al., 2022), natural language processing (Xu et al., 2020; Weld et al., 2022), medical image processing (Zhang et al., 2020), among many others (Ding et al., 2020; Yuan et al., 2020; Cheng et al., 2020; de Medrano & Aznarte, 2020).

## 3 SLOT-BASED TIME-SPACE TRANSFORMER WITH MEMORY BUFFER

To enhance the slot-based video models, e.g., SAVi and SAVi++, we introduce a new module called the *Slot-based Time-Space Transformer with Memory Buffer* (STATM) as the predictor. STATM is primarily designed to support causal reasoning and prediction for object-centric downstream tasks based on slots. This module consists of two key components: 1) the memory buffer, and 2) the Slot-based Time-Space Transformer (STAT). The memory buffer serves as a repository for storing historical slot information obtained from upstream modules, while STAT utilizes the information stored in the memory buffer for prediction and causal reasoning.

### 3.1 MEMORY BUFFER

The memory module is utilized for storing slot information from the upstream modules. We employ a queue-based storage mechanism. The representation of the memory buffer at time $t$ is given by:

$$M_t = \texttt{Queue}(S_i, \ldots, S_t), \tag{1}$$

where $S_t = \{s_{(0,t)}, \ldots, s_{(N,t)}\}$ represents the slot information extracted from the corrector module of SAVi and SAVi++ at time $t$. Here, $N$ signifies the number of slots, which is associated with the number of objects within the scene. The size of $M$ can be fixed or infinite. The new information is appended at the end of the queue.

### 3.2 SLOT-BASED TIME-SPACE TRANSFORMER (STAT)

The primary role of STAT lies in leveraging slot data from the memory buffer to facilitate slot-based dynamic temporal reasoning and spatial interaction computations. Furthermore, it integrates the outcomes of temporal reasoning and spatial interactions to achieve a unified understanding. Specifically, for temporal dynamic reasoning, a cross-attention mechanism is employed, which effectively utilizes historical context stored in the memory buffer to enable accurate predictions of future states. Meanwhile, for spatial interaction computations, we employ a self-attention mechanism that operates on slot representations to compute the relevance between different slots within the $S$. The results obtained from temporal dynamic reasoning and spatial interaction computation are merged to provide a holistic understanding encompassing both temporal dynamics and spatial interactions. This

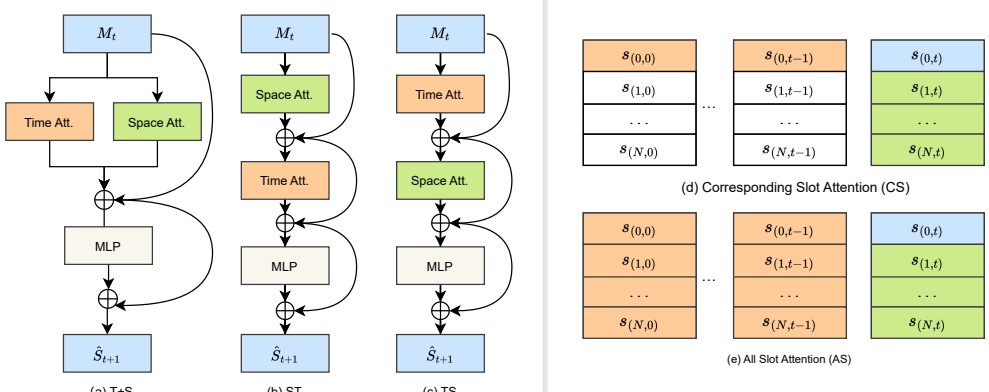

Figure 2: **Left:** Fusion approaches of spatiotemporal attention explored in our study. (a) The sum of computed temporal attention and spatial attention results (T+S). (b) Spatial attention computation followed by using the outcome as input for temporal attention (ST). (c) Temporal attention computation followed by using the outcome as input for spatial attention (TS). **Right:** Spatiotemporal attention computation architectures explored in our study. The green slots represent those employed for spatial attention computation, while the orange slots are indicative of those used for temporal attention computation. (d) Corresponding Slot Attention (CS). (e) All Slot Attention (AS).

comprehensive representation enhances the model's capability for accurate prediction and reasoning in object-centric tasks. We propose three approaches as illustrated in Figure 2a-c.

We also introduced two computational architectures for spatiotemporal attention as illustrated in Figure 2d-e. (1) *Corresponding Slot Attention (CS)*: For slot $s_{(i,t)}$, temporal attention is computed by using it and corresponding slots in $\{s_{(i,0)}, \ldots, s_{(i,t-1)}\}$, while spatial attention computation is performed using it and all slots within $\{s_{(0,t)}, \ldots, s_{(N,t)}\}$. (2) *All Slot Attention (AS)*: For slot $s_{(i,t)}$, temporal attention is computed by using it and all slots in $\{s_{(0,0)}, \ldots, s_{(N,t-1)}\}$. The spatial attention computation remains the same as in approach CS.

In the CS architecture, $s_{(i,t)}$ undergoes temporal attention computation exclusively with its corresponding slots. This design offers several notable advantages. Firstly, it enables a more robust association between objects and slots in terms of temporal sequences, preserving the slot's invariance with respect to the object. Additionally, this approach significantly reduces computational costs when compared to the AS structure. This efficiency makes the CS architecture an appealing choice for achieving effective temporal binding while optimizing computational resources.

In the AS architecture, the temporal attention involves calculating the attention between $s_{(i,t)}$ and all previous slots. The AS structure is designed to achieve improved slot-based prediction and reasoning in complex, unguided scenarios. The design rationale for AS is as follows. In previous time steps, objects were not effectively bound to specific slots, requiring each slot to search through memory to link relevant object information.

For example, when a person observes a car in a scene at time $t$ (assuming it was not noticed before), they often rely on their memory of previous scenes to determine where the car was previously located. This recall allows them to identify the previous position of this car and use it, along with the current one, to infer its future state. The AS architecture assumes that objects were not segmented in previous frames or that effective hints for segmentation were absent. In summary, if the upstream task effectively segments objects into slots, the CS architecture is preferred. Otherwise, the AS architecture can be considered.

For SAVi and SAVi++ models with hints in the first frame, the AS enhancement might not be significantly effective and could increase computational load. Since the predictor in both SAVi and SAVi++ is a transformer encoding block, all experiments and investigations in this paper only involve a STAT encoding block. We adopt the CS attention architecture with the T+S spatiotemporal fusion approach for our proposed STATM predictor. The memory buffer stores the slot information from the corrector for time steps. We then explain the calculation of spatiotemporal attention:

$$M_t = \texttt{Queue}\left(S_0, \ldots, S_t\right). \tag{2}$$

For a STAT encoding block, query/key/value vectors are computed for each slot:

$$k_{(i,t)}^a = LN_k^a\left(s_{(i,t)}\right) \in R^{D_h}, \quad q_{(i,t)}^a = LN_q^a\left(s_{(i,t)}\right) \in R^{D_h}, \quad v_{(i,t)}^a = LN_v^a\left(s_{(i,t)}\right) \in R^{D_h},$$
(3)

where $k$, $q$, and $v$ represent learned linear projections. $s_{(i,t)}$ denotes the vector of the $i$-th slot at time $t$. The latent dimensionality for each attention head is set to $D_h = D/A$.

The computation of spatiotemporal attention is also slot-based, and weights are calculated using dot-product. For the slot $s_{(i,t)}$, the spatiotemporal attention weights are computed as follows:

$$a_{(i,t)}^{(a)time} = \texttt{Softmax}\left(\left(\frac{q_{(i,t)}^{(a)}}{\sqrt{D_h}}\right)^{\mathsf{T}} \cdot \left[k_{(i,0)}^{(a)}\left\{k_{(i,t')}^{(a)}\right\}_{t'=0,\dots,T}\right]\right),$$

$$a_{(i,t)}^{(a)space} = \texttt{Softmax}\left(\left(\frac{q_{(i,t)}^{(a)}}{\sqrt{D_h}}\right)^{\mathsf{T}} \cdot \left[k_{(0,t)}^{(a)}\left\{k_{(i',t)}^{(a)}\right\}_{i'=0,\dots,N}\right]\right).$$
(4)

For each slot at time $t$, we calculate the weighted sum of value vectors using spatiotemporal attention coefficients from each attention head. The individual spatial or temporal attention systems in the CS structure can refer to Equation (5).

$$z_{(i,t)}^{(a)time} = \sum_{t'=0}^{T} a_{(i,t),(t')}^{(a)time} v_{(i,t')}^{(a)}, \qquad z_{(i,t)}^{(a)space} = \sum_{i'=0}^{N} a_{(i,t),(i')}^{(a)space} v_{(i',t)}^{(a)}.$$
(5)

The combined spatiotemporal vectors are individually linearly transformed, summed, and input into an MLP, where layer normalization (LN) is applied after each residual structure, v.i.z.,

$$s_{(i,t)}^{'time} = W_o^{time}\begin{bmatrix} z_{(i,t)}^{(1)time} \\ \vdots \\ z_{(i,t)}^{(A)time} \end{bmatrix} + s_{(i,t)}, \qquad s_{(i,t)}^{'space} = W_o^{space}\begin{bmatrix} z_{(i,t)}^{(1)space} \\ \vdots \\ z_{(i,t)}^{(A)space} \end{bmatrix} + s_{(i,t)},$$
(6)

$$s_{(i,t)}' = \texttt{LN}\left(s_{(i,t)}^{'time} + s_{(i,t)}^{'space}\right), \qquad \widehat{s}_{(i,t+1)} = \texttt{LN}\left(\texttt{MLP}\left(s_{(i,t)}'\right) + s_{(i,t)}'\right).$$
(7)

In this section, we focus on the computation process of the CS architecture using the T+S fusion approach for the STAT encoding block. In summary, temporal attention is calculated by jointly incorporating historical information from the memory buffer and spatial attention's slot $s_{(i,t)}$. Across all structural approaches, computations are based on slots, and the equation formulations remain consistent. Specific computation methods and procedures can be found in Figure 2.

## 4 EXPERIMENTS

The central aims of our experiments include: 1) To validate the efficacy of our model, incorporating STATM as a substitute for the transformer encoding block predictor within the SAVi and SAVi++ frameworks. 2) To investigate the effects of varying memory buffer sizes during both the training and inference stages on the performance of the model. 3) To assess the impact of different spatiotemporal methods integrated within STATM on the model's effectiveness.

**Metrics.** We selected the Adjusted Rand Index (ARI) (Rand, 1971; Hubert & Arabie, 1985) and the mean Intersection over Union (mIoU) as evaluation metrics. ARI quantifies the alignment between predicted and ground-truth segmentation masks. For scene decomposition assessment, we commonly employ FG-ARI, which is a permutation-invariant clustering similarity metric. It allows us to compare inferred segmentation masks to ground-truth masks while excluding background pixels. mIoU is a widely used segmentation metric that calculates the mean Intersection over Union values for different classes or objects in a segmentation task. It measures the overlap between the predicted segmentation masks and the ground-truth masks, indicating the quality of object segmentation. In the context of video analysis, mIoU is adapted to evaluate the consistency and accuracy of object segmentation and tracking across frames. It provides insights into how well the model captures the spatial relationships between objects in consecutive frames.

Table 1: Segmentation results on the MOVi dataset. All models were trained for 100k steps with a batch size of 32, which differs from the official implementation of SAVi (small, 100k steps, batch size of 64) and SAVi++ (500k steps, batch size of 64).

| Model | mIoU↑ (%) | | | | | FG-ARI↑ (%) | | | | |
|---|---|---|---|---|---|---|---|---|---|---|
| | A | B | C | D | E | A | B | C | D | E |
| SAVi | 62.8 | 41.6 | 22.0 | 6.8 | 4.0 | **91.1** | **70.2** | 50.4 | 18.4 | 10.8 |
| STATM-SAVi | **67.5** | **42.8** | **34.0** | **17.0** | **9.0** | **91.1** | 70.1 | **57.7** | **40.9** | **36.9** |
| SAVi++ | 82.8 | **52.5** | 47.8 | 43.6 | 26.1 | 96.7 | 78.5 | 76.3 | 81.5 | 81.7 |
| STATM-SAVi++ | **83.5** | **52.5** | **49.5** | **50.1** | **27.9** | **96.9** | **78.9** | **77.7** | **85.8** | **85.0** |

**Datasets.** To evaluate the performance of our model, we utilized the synthetic Multi-Object Video (MOVi) datasets (Research, 2020; Greff et al., 2022), the same datasets used for SAVi++ training. These datasets are divided into five distinct categories: A, B, C, D, and E. MOVi-A and B depict relatively straightforward scenes, each containing a maximum of 10 objects. MOVi-C, D, and E present more intricate scenarios with complex natural backgrounds. MOVi-C, generated using a stationary camera, presents scenes with up to 10 objects. Transitioning to MOVi-D, the dataset extends the object count to accommodate a maximum of 23 objects. Lastly, MOVi-E introduces an additional layer of complexity by incorporating random linear camera movements. Each video sequence is sampled at a rate of 12 frames per second, resulting in a total of 24 frames per second.

**Training Setup.** We conducted our experiments in JAX (Bradbury et al., 2018) using the Flax (Heek et al., 2020) neural network library. In all experiments except the ablation study in section 4.2, we used the STAT encoding block in combination with the CS attention architecture, featuring the T+S spatiotemporal fusion approach. For training the STATM-SAVi and SAVi models, we utilized videos comprising of 6 frames at a resolution of $64 \times 64$ pixels. The training process is conducted over 100,000 iterations. Similarly, the STATM-SAVi++ and SAVi++ models were trained on continuous videos consisting of 6 frames at a higher resolution of $128 \times 128$ pixels, with training duration encompassing 100,000 iterations. The batch size for training all models was set to 32. The buffer size was unconstrained during training, and the maximum length of effective information was limited to 6 due to the utilization of a 6-frame training sequence. The training process was executed on two A100 80GB GPUs, and bounding boxes were used as the conditioning for all models. The settings of other hyperparameters were consistent with those presented in SAVi and SAVi++.

## 4.1 IMPROVEMENT OF SAVI AND SAVI++ WITH STATM

To evaluate the STATM module, we chose: 1) using SAVi-small as the baseline model to compare the results of SAVi-small and STATM-SAVi; and 2) using SAVi++ as the baseline model to compare the results of SAVi++ and STATM SAVi++. Note that other baseline models that performed worse compared with SAVi (Kipf et al., 2021) and SAVi++ (Elsayed et al., 2022) were therefore not considered herein. The results are presented in Table 1.

It is observed that compared with SAVi and SAVi++, our model achieves higher mIoU and FG-ARI on the relatively simple MOVi-A and B datasets. As the dataset complexity increases, the advantages of our model become even more pronounced. We also conducted supplementary evaluations of our model, please refer to Appendix B. Clearly, utilizing STATM as the predictor significantly enhances the object tracking and segmentation capabilities of the slot-based video model, especially in complex scenarios. This also proves the importance and rationality of STATM, where slot-based temporal dynamic reasoning and spatial interactive computations combine to improve predictions, resulting in better object segmentation and slot alignment. Essentially, higher prediction accuracy leads to better segmentation performance. If predictions are highly accurate, we don't need to track objects at every step. Instead, we can focus on predicted locations, optimizing resource usage.

However, much like humans cannot predict the appearance of new objects in the next moment, the predictor faces similar limitations. At the initial moment, if the corrector cannot provide sufficiently accurate object information to the predictor, the predictor cannot offer precise prediction information for the corrector either. This situation leads to a vicious cycle, causing a gradual deterioration in the model's perceptual performance. When new objects appear, the model's performance drops dramatically (e.g., as seen in Figure 3 of the MOVi-D, when a new object emerges at $t = 5$, our

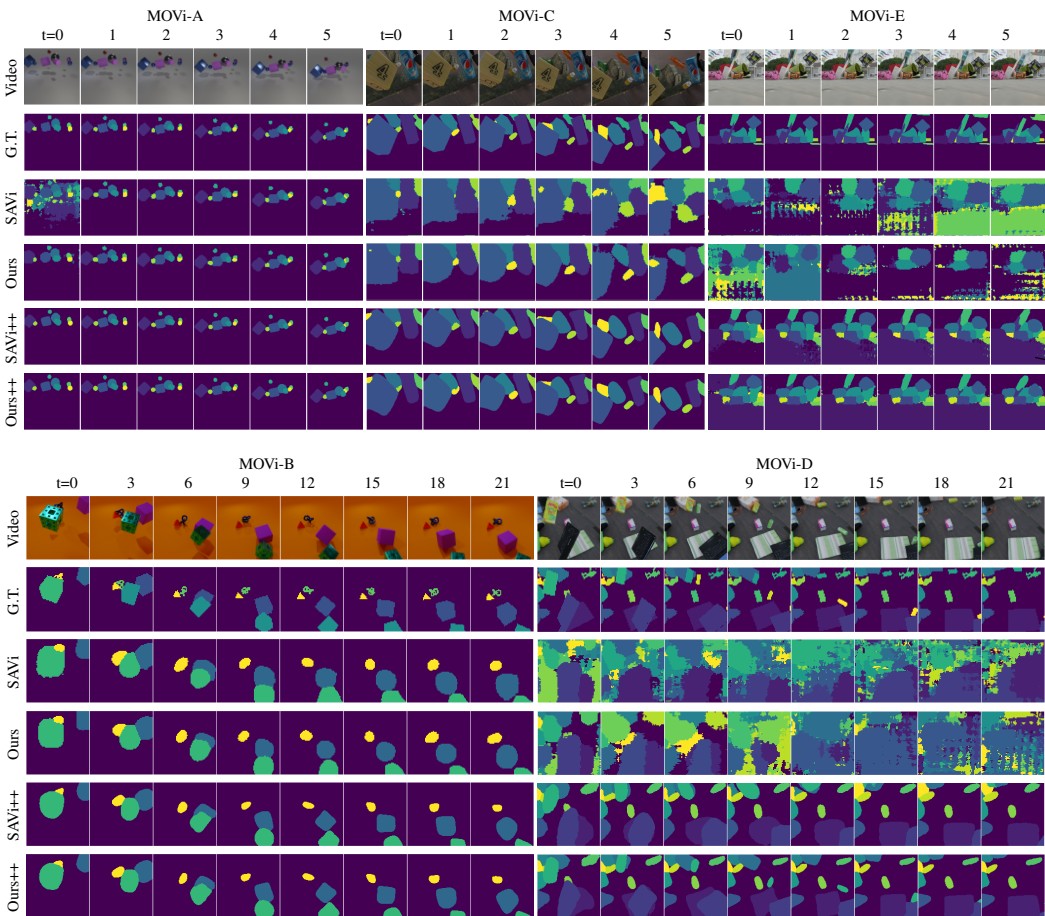

Figure 3: Qualitative results of our model compared to SAVi and SAVi++ on the MOVi dataset. Compared with SAVi and SAVi++, our model is slightly better than the SAVi/SAVi++ mode on the relatively simple MOVi-A and B data sets. However, as the complexity of the datasets increases, the advantage of our model becomes more pronounced.

model's segmentation quality deteriorates rapidly after that). In such cases, a simple predictor might even yield better results. This may also explain why using an MLP as a predictor in SAVi results in more stable training on complex datasets. If both the corrector and predictor are robust enough, this situation can be improved. The predictor can make accurate predictions based on the precise object information provided by the corrector, and the corrector can distinguish new objects from the predicted existing ones, thereby assigning new objects to separate slots.

Due to constraints of computing resources, our models were trained for 100k steps with a batch size of 32, which differs from the official implementation of SAVi (small, 100k steps, batch size of 64) and SAVi++ (500k steps, batch size of 64). Nevertheless, under equivalent conditions, our models consistently outperform the original counterparts: e.g., for the FG-ARI, STATM-SAVi (small, 100k steps, batch size of 32) achieves comparable performance to the official SAVi (large, 500k steps, batch size of 64) on MOVi-E datasets, while STATM-SAVi++ (100k steps, batch size of 32) performs comparably to SAVi++ (500k steps, batch size of 64). Importantly, the integration of a STAT encoding block does not lead to a significant increase in model parameters. Further improvements can be explored by increasing batch size and training steps, especially for STATM-SAVi++. We plan to investigate this in the future. A detailed comparison of parameters can be found in Appendix A.

## 4.2 ABLATION STUDY

In this section, we aim to evaluate the influence of different components of STATM, using STATM-SAVi as a baseline. Given the indispensability of the memory module for temporal attention, we

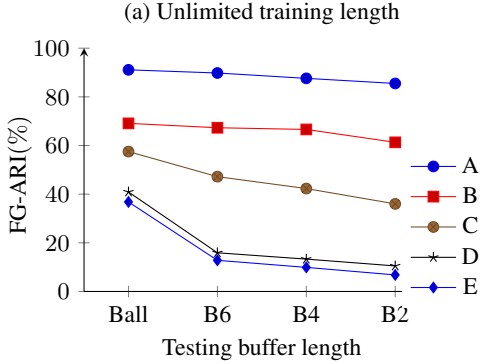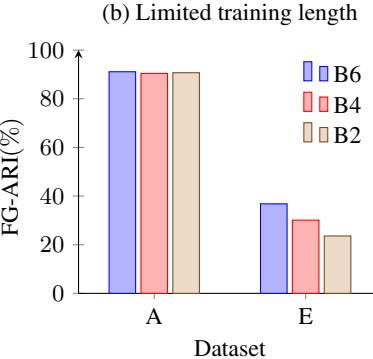

Figure 4: Segmentation results with different memory buffers on the MOVi dataset. (a) During training, all available information is utilized with an unlimited buffer length. The $x$-axis represents the model results during testing with buffer sizes ranging from unlimited to limited (6, 4, and 2 frames). (b) During training, the buffer length is limited to 6 frames. The different bars represent the model results during testing with buffer sizes of 6, 4, and 2 frames.

focus on two key aspects: 1) The effect of memory buffer size on the model during both training and inference phases; and 2) The influence of different spatiotemporal attention competition and fusion methods on the model.

**Ablation Experiment of Memory Module.** We have designed two sets of experiments to evaluate the impact of the memory buffer: 1) In the first set, we allowed an unlimited memory buffer length during training, but restricted it to a fixed length during testing, ensuring it didn't exceed the training buffer's length. To facilitate evaluation, we have not only assessed the model trained with 6 frames but also extended the training frames to 12, with the 12-frame results available in Appendix C. 2) In the second set, we fixed the buffer length during training, not exceeding the maximum buffer length, and removed any buffer length restrictions during testing. The results are shown in Figure 4.

Longer-duration video processing presents a challenge to the prediction and inference abilities of the model. It requires that the model extrapolate the learned physical laws of object motion to previously invisible segments. Therefore, the buffer's role during the testing becomes crucial for inference, especially for object tracking and segmentation beyond the training frame number (see Figure 4a). The prediction module requires additional information to summarize the physical laws of object motion, enabling it to make accurate predictions. This is similar to the human behavior.

Limiting the buffer length during the training phase reduces the segmentation and tracking capabilities of the model, but the decline is not overly serious. This aligns with human learning habits. Gathering more information at once is more conducive to humans in recognizing and summarizing patterns. However, when the overall learning duration remains constant, limitations in the field of view or learning content may lead to a decline in a person's ability to recognize and reason, but these abilities are not entirely lost. The model's tracking and segmentation capabilities over a duration equal to the training frames are less affected by memory (see Figure 4b). This is analogous to a scenario where a person has observed a significant amount of object motion in various scenarios over a time duration $t$. Subsequently, when asked to predict or describe how objects move within that $t$ time duration, as long as the inquiry doesn't extend beyond $t$, the person should still be able to provide reasonably accurate predictions and explanations, even if their view is obstructed or their memory is restricted. For more detailed analysis, please refer to Appendix C.

In summary, increasing the memory buffer size during both training and testing phases benefits the improvement of the model's perceptual capabilities across all datasets. However, for particularly complex datasets like MOVi-E, the excessive increase in the number of training frames may lead to a decline in the model's segmentation capabilities. In such cases, it might be worth considering improvements to modules like the encoder or corrector to enhance feature extraction capabilities.

**Ablation Experiments of Spatiotemporal Fusion and Computation.** We conducted ablation experiments of the spatiotemporal fusion method via the CS structure on the MOVi-A dataset. For

the ablation experiments related to the spatiotemporal computation structure, we chose the T+S fusion method. Since the AS structure was primarily designed for complex datasets, the computation method ablation experiments were conducted on the MOVi-E dataset. All models were trained using the first 6 frames of the video. The experimental results can be found in Table 2.

On the MOVi-E dataset, the segmentation capability of the AS structure is not as robust as that of the CS structure, but it's FG-ARI still outperforms the baseline. This suggests the following. 1) Compared to the transform encoding block, it produces more precise predictions for the STATM encoding block with the AS structure as the predictor, enhancing the object segmentation

Table 2: Result of spatiotemporal fusion method.

| Model | mIoU↑ (%) | | FG-ARI↑ (%) | |
|---|---|---|---|---|
| | A | E | A | E |
| STATM (CS, ST) | 58.4 | - | 90.9 | - |
| STATM (CS, TS) | 61.2 | - | 89.7 | - |
| STATM (CS, T+S) | 67.5 | 8.5 | 91.1 | 36.8 |
| STATM (AS, T+S) | - | 3.8 | - | 12.2 |

and tracking abilities of slot-based models like SAVi in complex video scenes. 2) As mentioned earlier, the AS structure is designed to handle scenes where objects are not effectively segmented into corresponding slots. Appendix C indicates that with the assistance of initial frame cues, SAVi exhibits decent scene decomposition in the early frames of test videos from the MOVi-E dataset. However, as time progresses, the lack of dynamic temporal interactions among corresponding slots and the impact of complex backgrounds lead to declining segmentation and tracking performance. Currently, models without prompts have limited relevance to our objectives. Hence, we choose not to conduct extensive experiments to verify the capabilities of the STATM with the AS structure.

## 4.3 LIMITATIONS

We used STATM as a prediction module to enhance the perceptual capabilities of slot-based models like SAVi and SAVi++. However, we didn't assess our model using real-world datasets. The foundation of our model's construction is based on the principle that "prediction and correction mutually reinforce each other". However, our evaluation of the rationality and effectiveness of STATM is based on the experimental results from the correction step, and we haven't directly tested its physical learning and reasoning abilities. This remains a significant focus for our future research. In this article, we didn't explore models with unconditional prompts. Verifying and improving the effectiveness of STATM models with different structures under unconditional prompts will be one of our main tasks in the future. In addition, the relationship between the model and humans is currently explained and analyzed from a rationality perspective. In the future, we intend to further optimize and improve our models by incorporating expertise from other domains (e.g., brain science). We will continue to explore the connection between deep learning, human causal reasoning, as well as intuitive physics.

## 5 CONCLUSION

In the real world, all objects follow the laws of physics. Intuitive physics serves as the bridge and connection through which humans comprehend the world. Our research aims to construct biologically plausible deep learning models to explore whether deep learning models can learn physical concepts like humans, and use these learned physical laws to make inferences and predictions about the future motion of objects.

We have designed a more reasonable prediction module called STATM, which clearly improved SAVi and SAVi++ models in the context of scene understanding and prediction. We demonstrated that reasoning and prediction abilities influence the model's scene object segmentation and tracking. The more accurate the reasoning and prediction abilities, the stronger the segmentation and tracking of objects. Through a series of experiments, we investigated the influence of memory and spatiotemporal reasoning on the model's perceptual abilities. We also attempted to provide reasonable explanations, which hold importance for the present interdisciplinary research across fields of AI and brain science. Although there still remain many challenges on this topic, the results in this paper illustrate that well-designed deep learning models can mimic human perception. Yet, in the future, we will continue exploring more cognitive theories as a basis, further improving and optimizing our model.

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

**APPENDIX**

## A   ADDITIONAL PARAMETER

Using the STTM structure as a predictor does lead to a slight increase in the parameter count of the SAVi and SAVi++ models. However, under the same training settings, our model achieves superior metrics. This suggests that the moderate increase in the parameter count doesn't significantly increase the training complexity of our model.

STATM-SAVi-Small indeed has a parameter increase of approximately 66K compared to SAVi-Small, which is notably smaller than the parameter increase seen in SAVi-Large compared to SAVi-Small (around 21378K parameters). Moreover, our STATM-SAVi-Small model, trained for 100k steps with a batch size of 32, performs similarly to the official SAVi-Large model, trained for 5000k steps with a batch size of 64. This further highlights the reasonableness and superiority of our designed prediction module.

## B   ADDITIONAL SEGMENTATION RESULTS

In order to better assess our model, we conducted an evaluation using the first 6 frames of the videos. Referring to Tables 1 in the main text, we can observe the following trends: on simple datasets, the decline in our model's object segmentation and tracking capabilities over extended time sequences is comparable to that of the baseline model. However, on complex datasets like MOVi-C, D, and E, the decrease in our model's performance is significantly less than that of the baseline model. This indicates that the STATM is more suitable for handling object segmentation and tracking tasks in longer-time sequences and complex environments. This finding further validates the effectiveness of our STATM model, as it draws inspiration from how humans track and segment objects in the real world.

## C   ADDITIONAL ABLATION EXPERIMENT OF MEMORY MODULE

**Training with unlimited buffer length and testing with limited buffer length.** To better assess the impact of the buffer on the model, we trained the model using the first 12 video frames, as shown in Table A.3 and A.4. We observed that: 1) On relatively simple datasets like MOVi-A, B, C, and D dataset, increasing the amount of training data with additional information would lead to a stronger

Table A.1: Comparison of the parameter number for different models.

| Model | Parameter Number | Model | Parameter Number |
|---|---|---|---|
| SAVi-Small | 895,268 | STATM-SAVi-Small | 961,572 |
| SAVi-Medium | 1,140,740 | STATM-SAVi-Medium | 1,207,044 |
| SAVi-Large | 22,273,412 | STATM-SAVi-Large | 22,339,716 |
| SAVi++ | 23,132,165 | STATM-SAVi++ | 23,264,389 |

Table A.2: Segmentation results on the first 6 frames of the MOVi dataset.

| Model | mIoU↑ (%) | | | | | FG-ARI↑ (%) | | | | |
|---|---|---|---|---|---|---|---|---|---|---|
| | A | B | C | D | E | A | B | C | D | E |
| SAVi | 66.9 | 49.3 | 29.7 | 13.9 | 8.3 | 92.3 | 80.1 | 69.2 | 45.5 | 32.2 |
| STATM-SAVi | 71.0 | 51.6 | 43.5 | 21.9 | 12.5 | 92.6 | 81.7 | 73.0 | 50.2 | 54.7 |
| SAVi++ | 85.2 | 59.5 | 55.3 | 49.8 | 30.7 | 97.2 | 86.3 | 83.9 | 87.1 | 88.2 |
| STATM-SAVi++ | 85.8 | 59.8 | 56.8 | 56.7 | 31.1 | 97.2 | 86.6 | 83.9 | 89.2 | 88.6 |

Table A.3: Evaluation on all video frames of the model trained using 12 frames (B represents the size of the buffer during the testing phase).

| Model | mIoU↑ (%) | | | | | FG-ARI↑ (%) | | | | |
|---|---|---|---|---|---|---|---|---|---|---|
| | A | B | C | D | E | A | B | C | D | E |
| STATM (Ball) | 66.9 | 39.3 | 26.1 | 13.8 | 4.3 | 92.3 | 72.9 | 62.5 | 59.6 | 17.9 |
| STATM (B12) | 66.2 | 39.3 | 25.9 | 13.2 | 3.9 | 91.3 | 73.0 | 60.8 | 55.6 | 10.4 |
| STATM (B6) | 64.3 | 39.3 | 25.4 | 12.3 | 3.6 | 89.3 | 72.7 | 57.4 | 50.6 | 5.6 |
| STATM (B4) | 62.8 | 39.1 | 24.8 | 11.8 | 3.4 | 88.4 | 72.5 | 55.1 | 47.7 | 4.4 |
| STATM (B2) | 59.1 | 38.2 | 23.9 | 11.2 | 3.1 | 85.5 | 70.6 | 51.1 | 44.0 | 3.5 |

Table A.4: Evaluation on the first 6 video frames of the models trained by 6 frames and 12 frames (T represents the number of frames used for training model, B represents the size of the buffer during the testing phase).

| Model | mIoU↑ (%) | | | | | FG-ARI↑ (%) | | | | |
|---|---|---|---|---|---|---|---|---|---|---|
| | A | B | C | D | E | A | B | C | D | E |
| STATM (T6, Ball) | 71.0 | 51.6 | 43.5 | 21.9 | 12.5 | 92.6 | 81.7 | 73.0 | 50.2 | 54.7 |
| STATM (T6, B6) | 71.0 | 51.6 | 43.5 | 21.9 | 12.5 | 92.6 | 81.7 | 73.0 | 50.2 | 54.7 |
| STATM (T6, B4) | 71.0 | 51.3 | 42.7 | 19.7 | 12.0 | 92.6 | 81.7 | 72.5 | 46.8 | 51.0 |
| STATM (T6, B2) | 70.8 | 49.2 | 38.7 | 14.6 | 10.2 | 91.8 | 80.6 | 69.1 | 34.6 | 37.3 |
| STATM (T12, Ball) | 60.2 | 42.7 | 28.1 | 15.4 | 7.6 | 92.7 | 82.9 | 73.6 | 55.5 | 33.5 |
| STATM (T12, B12) | 60.2 | 42.7 | 28.1 | 15.4 | 7.6 | 92.7 | 82.9 | 73.6 | 55.5 | 33.5 |
| STATM (T12, B6) | 60.2 | 42.7 | 28.1 | 15.4 | 7.6 | 92.7 | 82.9 | 73.6 | 55.5 | 33.5 |
| STATM (T12, B4) | 59.7 | 42.8 | 28.0 | 15.4 | 7.3 | 92.1 | 82.9 | 73.3 | 54.5 | 29.3 |
| STATM (T12, B2) | 58.3 | 42.5 | 27.7 | 14.9 | 6.4 | 89.6 | 82.5 | 71.6 | 50.2 | 19.4 |

model, as expected. 2) On the MOVi-E dataset, increasing the number of training frames resulted in a decrease in the model's tracking and segmentation capabilities. This could be attributed to the limitations in the ability of the upstream modules to effectively extract image features. The findings from the SAVi and SAVi+, which used more powerful encoders and data augmentation to improve segmentation performance on MOVi-E, support this observation. Therefore, exploring the design of a more robust encoder and refining the corrector and guidance modules may yield unexpected improvements. We plan to further investigate this direction in future research.

**Training with limited buffer length and testing with limited buffer length.** During our research, we developed an intriguing idea that aligns to some extent with human behavior: In the process of learning and memory, individuals need to observe and summarize the motion patterns of objects. They can then use these learned physical principles to predict the motion of objects in various scenarios. Two conditions can significantly influence an individual's learning outcomes and adaptability:

1) If individuals are not constrained by the field of vision during the learning phase, meaning they can observe the complete motion trajectory of objects at once, they may establish a more extensive and comprehensive understanding of physical principles. Consequently, they may have higher expectations for scenarios in the testing phase where there are no vision constraints. However, during the testing phase, suddenly restricting their field of vision within a certain range might lead to confusion and discomfort because they have become accustomed to a broader perspective.

2) Conversely, if individuals are subject to the field of vision constraints during the learning phase, meaning they can only observe partial motion trajectories of objects, they may have already adapted to this restricted condition and developed corresponding knowledge of physical principles. There-

Table A.5: Evaluation result of the model trained limited buffer length (T represents the size of the buffer during the training phase, B represents the size of the buffer during the testing phase).

| Model | mIoU↑ (%) | | | | FG-ARI↑ (%) | | | |
|---|---|---|---|---|---|---|---|---|
| | First 6 frames | | All frames | | First 6 frames | | All frames | |
| | A | E | A | E | A | E | A | E |
| STATM (T2, Ball) | 71.6 | 9.9 | 66.9 | 6.2 | 92.6 | 41.2 | 90.7 | 23.6 |
| STATM (T2, B6) | 71.6 | 9.9 | 69.0 | 5.7 | 92.6 | 41.2 | 91.5 | 22.7 |
| STATM (T2, B4) | 71.7 | 9.9 | 69.3 | 5.4 | 92.6 | 41.5 | 91.2 | 20.0 |
| STATM (T2, B2) | 71.9 | 9.5 | 69.5 | 4.8 | 92.6 | 41.3 | 91.2 | 15.5 |
| STATM (T4, Ball) | 73.6 | 9.8 | 68.0 | 6.8 | 92.5 | 41.7 | 90.4 | 30.1 |
| STATM (T4, B6) | 73.6 | 9.8 | 69.5 | 4.7 | 92.5 | 41.7 | 90.3 | 15.1 |
| STATM (T4, B4) | 73.7 | 9.7 | 69.6 | 4.3 | 92.4 | 41.3 | 90.1 | 11.8 |
| STATM (T4, B2) | 74.0 | 9.0 | 69.3 | 3.9 | 92.1 | 38.4 | 89.2 | 9.1 |
| STATM (T6, Ball) | 71.0 | 12.5 | 67.5 | 8.5 | 92.6 | 54.7 | 91.1 | 36.8 |
| STATM (T6, B6) | 71.0 | 12.5 | 66.1 | 5.4 | 92.6 | 54.7 | 89.8 | 12.8 |
| STATM (T6, B4) | 71.0 | 12.0 | 64.2 | 4.9 | 92.6 | 51.0 | 87.6 | 9.9 |
| STATM (T6, B2) | 70.8 | 10.2 | 61.6 | 4.2 | 91.8 | 37.3 | 85.5 | 6.8 |

fore, in the testing phase, under similar vision constraints, they might perform relatively better because their learning background has already adapted to these limitations.

To examine whether our model exhibits these characteristics, we intentionally limited the buffer size during both the training and testing phases, and the model evaluation results are presented in Table A.5. Remarkably, we found that the model trained with a smaller buffer (equivalent to restricted vision) experienced less impact from the buffer during the testing phase. This suggests that our model aligns to some extent with human behavior. This has intriguing implications for the fusion of deep learning and cognitive science. However, it's important to note that real human learning and cognitive processes are likely more complex and influenced by various factors. This study provides a theoretical framework, but further theoretical substantiation and experimental validation are still needed.

