# OpenReview forum: "Reasoning-Enhanced Object-Centric Learning for Videos"
_ICLR.cc/2024/Conference — ICLR 2024 Conference Withdrawn Submission_

### Official Review · Reviewer_vpNg · 2023-10-14

**Soundness:** 2 fair
**Presentation:** 3 good
**Contribution:** 1 poor
**Rating:** 3
**Confidence:** 5

**Summary:**

This paper proposes a memory-enhanced Predictor in slot-based video object-centric learning (OCL) frameworks. The proposed Predictor, STATM, runs self-attention over slots at the same timestep and across timesteps. Experimental results on MOVi datasets show that STATM serves as a plug-and-play module to consistently improve the segmentation results of SAVi and SAVi++.

**Strengths:**

- The Predictor is indeed an under-explored component in SAVi-like models. This paper serves a preliminary attempt in this direction
- The ablation studies are thorough

**Weaknesses:**

My biggest concern is regarding the experimental settings:
- The reported performances of baselines are not with their best training configs. The reason "computation constraint" is not acceptable, as I am not sure with longer training & larger batch size, will the performance gain disappear
- As a result, the performance of STATM is not SOTA. SAVi++ reports a mIoU of 47.1 on MOVi-E, which is much higher than this paper
- Only comparing with SAVi and SAVi++ is also not enough. There have been several follow-up works since then, such as STEVE [1]. STEVE does not use the initial frame hint, but still produces meaningful segmentation results on MOVi datasets. I believe with initial frame hints, STEVE will even outperform SAVi++. So this baseline should be considered as well
- The experiments in the paper mainly focus on object segmentation. While it is an important outcome of OCL, the quality of learned object slots is another important aspect. I would suggest the authors to at least perform an object property prediction experiment on MOVi, e.g., follow the protocol in LSD [1]

[1] Singh, Gautam, Yi-Fu Wu, and Sungjin Ahn. "Simple unsupervised object-centric learning for complex and naturalistic videos." NeurIPS, 2022.

[2] Jiang, Jindong, et al. "Object-centric slot diffusion." arXiv preprint arXiv:2303.10834 (2023).

**Questions:**

Besides the questions raised in Weaknesses. I have a few minor questions:
- What is the computation and memory overhead STATM brings compared to SAVi and SAVi++? Please report the training memory and speed with and without STATM
- I agree that the current MLP or Transformer Predictor in SAVi/SAVi++ does not consider long-term temporal information. However, will adding an LSTM after the Predictor fix this? Like a Transformer-LSTM module. I wonder if the authors have tried this variant

---

> ### Author Response · Authors · 2023-11-11
>
> Thank you for your review comments. I will provide answers to the questions you raised.
> 1. Regarding weaknesses 1 and 2.  During our subsequent work, we realized that people might have questions about the results for the MOVi-E dataset. Therefore, we additionally trained the STATM-SAVi++ module on the MOVi-E dataset with batch_size = 64 and training_steps = 500k (consistent with the settings in the SAVi++ paper). The results are mIoU=57.0 and FG-ARI=89.2, which exceeds the results reported in the SAVi++ paper( mIoU=47.1 and FG-ARI=84.1) and github (https://github.com/google-research/slot-attention-video, FG-ARI=85.1). This indicates that with longer training and a larger batch size, the model's gains do not diminish. It is also evident that the model's results are much higher than those reported in the SAVi++ paper. We will include these results in the supplementary material.
> 2. Regarding weaknesses 3. First of all, I'm sorry that I may not know exactly what you mean. We conducted quantitative experiments in the paper and compared the experimental results to show the advantages of our model, but you guessed that our model may lose gain with longer training & larger batch size.
> However, for the STEVE model, it has only been compared with SAVi-Large model, and its FG-ARI on long sequences in MOVi-E is approximately between 50 and 55, which is significantly lower than SAVi++(FG-ARI=84.1).  The STEVE paper also does not provide the mIoU metric. As for the hinted version you mentioned, we have not found relevant information, yet you believe that it might surpass SAVi++.
> Your behavior in this regard is unfair to us. I'm sorry, but currently, we cannot draw the same speculative conclusion as you from the STEVE paper. If their paper later provides a hinted version of STEVE, we would be happy to make a comparison. Respected reviewer, please kindly reconsider this suggestion to avoid any confusion or inconvenience.
>
> 3. Regarding weaknesses 4. The quality of learning object slots is indeed an important aspect.  This work will be placed in follow-up work on whether the model can learn general physical laws and use physical laws to make predictions. We plan to open source the code and share the results once follow-up work is completed.
> 4. Regarding the question 1.
> We intentionally set the batch size very large and received the following out-of-memory error message:
> (1) SAVi++:
> RuntimeError:
> RESOURCE_EXHAUSTED: Out of memory while trying to allocate 106793918792 bytes.
> BufferAssignment OOM Debugging.
> BufferAssignment stats:
> parameter allocation: 363.80MiB.
> constant allocation: 45B.
> maybe_live_out allocation:  264.73MiB.
> preallocated temp allocation:   99.46GiB.
> preallocated temp fragmentation:  295.48MiB (0.29%).
> total allocation:   99.81GiB.
> total fragmentation:  295.55MiB (0.29%).
> (2) Ours++:
> RuntimeError:
> RESOURCE_EXHAUSTED: Out of memory while trying to allocate 106773606256 bytes.
> BufferAssignment OOM Debugging.
> BufferAssignment stats:
> parameter allocation: 365.31MiB.
> constant allocation: 100B.
> maybe_live_out allocation:  266.25MiB.
> preallocated temp allocation:   99.44GiB.
> preallocated temp fragmentation:  295.48MiB (0.29%).
> total allocation:   99.80GiB.
> total fragmentation:  295.56MiB (0.29%).
> (3) The speed on one A100 (batch_size=8):
>  SAVi++ : Setting work unit notes: 0.5 steps/s, 0.0% (49/500000), logging_writer.py:48] [49] steps_per_sec=0.549965, logging_writer.py:48] [82] steps_per_sec=0.549036.
> STATM-SAVi++: Setting work unit notes: 0.5 steps/s, 0.0% (40/500000), logging_writer.py:48] [40] steps_per_sec=0.548861, logging_writer.py:48] [73] steps_per_sec=0.547787.
> 5. Regarding the question of LSTM, we did consider it at some point. However, similar to the first question you raised, we took into account that it might lead to an increase in parameters and a decrease in training speed. Therefore, in our paper, we did not try this structure.
>
> Thank you once again for your valuable feedback. If you have further questions, we'd be happy to discuss them with you.

---

> > ### Comment · Reviewer_vpNg · 2023-11-16
> > **Re: rebuttal**
> >
> > 2. I understand STEVE does not have a hinted version. However, I think you should try to combine STATM with the unhinted version of STEVE to see if it can improve performance. Because SAVi/SAVi++ only represents one line of OCL works which uses mixture-based slot decoder, while the Transformer-based slot decoder in STEVE has shown impressive results these years. This is an important experiment to test the generality of your method.
> > 3. It is not acceptable to say "leave it for future work" as evaluating the quality of slots is itself an important part of OCL. As I suggested, you can do a simple object property prediction task as done in LSD or Slot Attention, i.e. attaching MLP heads to the slots and learning to predict the object's color, position, etc.
> > 4. I am not asking you to increase the batch size. What I mean is under the same batch size settings, what is the memory and training time of STATM compared to SAVi? Does your method consume more memory?

---

> ### Author Response · Authors · 2023-11-17
>
> 2. ***At present, our goal is to obtain a model that performs well in both prediction and perception, even if it requires a small amount of supervision and cues. This represents a significant advancement for us in combining intuitive physics with artificial intelligence. We need a model that can learn and utilize universal laws of physics for predictions. It is crucial for the model's perception abilities to also perform well, as we have found that perception and reasoning in human daily behavior are mutually influential. For instance, if there is a cat hidden in dense foliage, it might be challenging for us to detect it. Yet, if someone points out the cat's location, we can identify it. However, considering that the cat might be in motion, jumping onto a tree or into a thicket, this poses challenges for continued tracking and identification. Yet, if an individual can effectively understand the cat's movements (temporal reasoning) and anticipate its potential hiding places (spatial interaction), they can more easily continue identifying and tracking the cat.
> ***Unfortunately, STEVE does not meet our requirements. We have not neglected experiments on STEVE, after conducting comparative experiments, we observed that STEVE has weaker predictive capabilities, and its perception performance in certain scenarios is inferior to SAVi++. Therefore, STEVE lacks significant value for our research, as its predictive and perceptual abilities do not mutually enhance each other.
> *** In the field of artificial intelligence for science, Google's models typically emphasize both accuracy and rationality, maintaining an overall balance in the models. However, when others seek to improve these models, there is often a tendency to excessively pursue accuracy at the expense of other aspects of performance. Using STEVE as an example, its video reconstruction or prediction capabilities tend to weaken in such optimization processes.
> ***Our pursuit is to enhance accuracy while concurrently incorporating more rationality, which is crucial for our research in intuitive physics and artificial intelligence.
> 3. No problem. We have conducted slot evaluation experiments, such as visualizing the attention of each slot to objects over time. If needed, we can also perform separate decoding for each slot.
> 4. ***Relative to SAVi++，STATM hardly reduces training speed (one A100 , MOVi-E).
> SAVi++ :steps_per_secconds=0.549036. Ours++ :steps_per_secconds=0.547787.
> ***Relative to SAVi++, STATM incurs a slight increase in memory consumption (about 0.14GB, one A100, same batch size, MOVi-E).
> SAVi++: about 73841MiB. Our++: about 73987MiB.

---

> > ### Comment · Reviewer_vpNg · 2023-11-21
> > **Re: Rebuttal**
> >
> > While the authors emphasize that reasoning is their core focus in the paper, they do not have any experiments showing their learned slot model applied to downstream reasoning tasks, like VQA (e.g. as done in SlotFormer [1]). I do not think this is a valid claim and thus maintain my score of rejection.
> >
> > [1] Wu, Ziyi, et al. "Slotformer: Unsupervised visual dynamics simulation with object-centric models." arXiv preprint arXiv:2210.05861 (2022).

---

> > > ### Author Response · Authors · 2023-11-23
> > > **Thanks for your reply**
> > >
> > > We would like to thank you for your picky reply. While part of your comments and suggestions are valuable, we hold completely different opinions on other picky ones. Anyway, thanks for your time and efforts placed on reviewing our paper. We will withdraw the paper and revise it for submission to another avenue.

---

### Official Review · Reviewer_zBsm · 2023-10-31

**Soundness:** 2 fair
**Presentation:** 2 fair
**Contribution:** 2 fair
**Rating:** 3
**Confidence:** 4

**Summary:**

This paper introduces the Slot-based Time-Space Transformer with Memory buffer (STATM), an object-centric learning model for videos. The model replaces the predictor in SAVi and SAVi++ with a spatiotemporal attention component that attends to slots in previous timesteps, which are stored in a memory buffer. The model is evaluated on the MOVi datasets, showing improvements in segmentation quality.

**Strengths:**

This paper tackles an important problem of improving object-centric learning models for videos. While spatiotemporal transformers have been applied to Slot Attention-based video models before [1], they have not been trained in an end-to-end fashion, as far as I know. The experimental results show improvements over SAVi and SAVi++, especially on the more complex MOVi-D and MOVi-E datasets, although I have concerns about these results that I state below.

[1] SlotFormer: Unsupervised Visual Dynamics Simulation with Object-Centric Models. https://arxiv.org/abs/2210.05861

**Weaknesses:**

There a few instances in the paper where the authors make statements that are not well-supported by their experiments. For example,

- The title and abstract emphasize reasoning, but this is not supported by any experiments. I think it is fine to use reasoning as a motivation for their model, but if the title includes “Reasoning-Enhanced”, I would have expected some experiments showing this ability.
- Similarly, I feel several statements connecting their model to human behavior are too strong and not supported by empirical evidence. For example, in the abstract, they state “We demonstrated that the improved deep learning model exhibits certain degree of rationality imitating human behavior”, but I don’t believe this is strongly supported by their experiments.
- In Section 4.1, “Essentially, higher prediction accuracy leads to better segmentation performance. If predictions are highly accurate, we don’t need to track objects at every step. Instead, we can focus on predicted locations, optimizing resource usage.” While the results do show STATM to perform better than SAVi, it is not clear that this is because of more accurate predictions. There is also no evidence of improved resource usage from STATM.

I also have a few issues with the experimental results section:

- The results for SAVi on MOVi-D and MOVi-E seem very low when compared to the SAVI++ paper (18.4 vs. 59.6 FG-ARI for MOVi-D and 10.8 vs. 55.3 FG-ARI for MOVi-E). I understand the authors only trained for 100k steps with a smaller batch size, but this indicates to me that the model is not well-trained yet. While we can draw some conclusions about the training speed from these results, I think we need to be careful about drawing broader conclusions, especially when the converged value (from the SAVI++ paper) is so much higher.
- The first ablation where we test on a smaller buffer length than we train on does not seem too informative. In this case, the model is being trained on a different setting than it was tested on so we can expect the performance to decrease.


Minor typos in Figure 1:

- Hits → Hints
- Preceive → Perceive
- Memery → Memory

**Questions:**

- Is there any positional encoding applied to the slots (either time-wise or slot-wise or both)? Otherwise, how are slots from different timesteps distinguished?
- In the introduction, the authors write “The prediction step in SAVi and SAVi++ is similar to human inference, but the predictor module in SAVi and SAVi++ is somewhat simplistic, as it relies solely on single-frame information from the current time step for prediction.” While it is true that the predictor in SAVi only uses the slots from the current time step, the slots themselves may contain information from previous timesteps (eg. velocity) since they are updated iteratively. One thing I would be curious about is if the representations in STATM differ from the representations in SAVi in that they may not need to include information such as velocity since this can be inferred by the spatiotemporal transformer. One way to test this would be to try to predict velocity from the slot representations.
- In Equation 4, I am a bit confused about the notation. Why are k_{i, 0} and k_{0, t} written separately outside the brackets?
- At the end of Section 4.2, the authors write "Currently, models without prompts have limited relevance to our objectives.” Prompting is also mentioned in the limitations section, but not discussed elsewhere in the paper. What is prompting in this context?
- For clarity, I would suggest only explaining T+S and CS in the main text since that is what is used in most of the experiments. The explanation of the other variants can go in the ablations section.
- This model seems like it would be especially beneficial in cases where an object that appears in the first frame is fully occluded at some intermediate frame and then reappears in a later frame. I think it would be informative to try some experiments exploring this (even in toy settings).

---

> ### Author Response · Authors · 2023-11-11
>
> Thank you for your review comments. I will provide answers to the questions you raised.
> 1. Regarding the weakness 1.1. Our initial intention was to create a model that emulates human-like behavior, capable of learning universal physical knowledge and utilizing it for predictions. However, during the prediction process, we realized that human reasoning plays a significant role in the perception of complex scenes. Reasoning and perception are distinct aspects of the work, the predictive part is a task we plan to undertake in subsequent work. Including it in this paper could lead to excessive length.
> 2. Regarding the weakness 1.2. All our explanations are based on common sense from a human perspective. At the current stage, the focus of brain-like science and intuitive physics is inherently a human-centric area of study. Many phenomena can only be explained from the standpoint of common sense and whether they align with human behavior.
> 3. Regarding the weakness 1.3. Taking SAVi++ as an example, our model achieves results comparable to or even superior to those mentioned in the SAVi++ paper on all datasets outside of MOVi-E, using much fewer training steps and batch sizes than SAVi++. This confirms the conclusion of optimized resource utilization in terms of computational resources and efficiency.
> 4. Regarding the weakness 2.1 of experimental results, our paper is explicit about the comparison between SAVi-small and STATM-SAVi-small. Please refer to GitHub (<https://github.com/google-research/slot-attention-video>) where the results for SAVi-small  are provided as FG-ARI 33.8$\pm$7.7 for MOVi-D and  FG-ARI 8.3$\pm$0.9 for MOVi-E. You can clearly see on github that the results mentioned in the SAVi++ paper are for the SAVi-large model. I kindly request you to review this information to avoid any confusion or misunderstanding about the experimental results.
> During our subsequent work, we realized that people might have questions about the results for the MOVi-E dataset. Therefore, we additionally trained the STATM-SAVi++ module on the MOVi-E dataset with batch_size = 64 and training_steps = 500k (consistent with the settings in the SAVi++ paper). The results are mIoU=57.0 and FG-ARI=89.2, which exceeds the results reported in the SAVi++ paper( mIoU=47.1 and FG-ARI=84.1) and github (https://github.com/google-research/slot-attention-video, FG-ARI=85.1). This indicates that with longer training and a larger batch size, the model's gains do not diminish. It is also evident that the model's results are much higher than those reported in the SAVi++ paper. We will include these results in the supplementary material.
> 5. Regarding the weakness 2.2 of the question "The first ablation where we test on a smaller buffer length than we train on does not seem too informative," our ablation experiments in the testing phase were conducted not only on a smaller buffer than the training set but also on a larger buffer, for example, 24 frames.
> 6. Regarding question 1, there is positional encoding, and we use the default indices of the buffer for its encoding.
> 7.  Regarding question 2, just like our response 1, prediction is our next stage task.
> 8.  Regarding question 3, the purpose of doing this is to provide an example for better understanding.
> 9.  Regarding question 4, in **Training Setup** of Section 4, we mentioned that the prompt information used in this paper is the bounding box of the first frame. However, the prompt information can also be the segmentation of the first frame.
> 10. Regarding question 5, in the **4.2 Ablation Study**, we mainly conducted experiments, while in the model section, we primarily introduced the main model (CS) used in this paper. Apart from CS, we did not spend much space introducing other models, only showcasing them in the Figure 2.
> 11. Regarding question 6, your point is very valid, and indeed, we have explored this in subsequent prediction work. We will soon publish the results.
> 12. Furthermore, other spelling errors will be corrected and re-uploaded after the revisions.
>
> Thank you once again for your valuable feedback. If you have further questions, we'd be happy to discuss them with you.

---

> > ### Comment · Reviewer_zBsm · 2023-11-21
> > **Response to Rebuttal**
> >
> > Thank you for taking the time to address my concerns and correcting my incorrect comparison with SAVi-large. The new MOVi-E results run to 500k are encouraging. Regarding 8, should t’ then go from 1 to T and i’ from 1 to N?
> >
> > Overall, I still do feel the connection of the model to human behavior and reasoning are too strong and not supported by experiments or citations. This concern is shared by reviewer km72. After reading the responses as well as the other reviews, I will maintain my original score.

---

> > > ### Author Response · Authors · 2023-11-23
> > > **Thanks for your reply**
> > >
> > > We would like to thank you for your picky reply. While part of your comments and suggestions are valuable, we hold completely different opinions on other picky ones. Anyway, thanks for your time and efforts placed on reviewing our paper. We will withdraw the paper and revise it for submission to another avenue.

---

### Official Review · Reviewer_4XrW · 2023-10-31

**Soundness:** 2 fair
**Presentation:** 2 fair
**Contribution:** 2 fair
**Rating:** 5
**Confidence:** 5

**Summary:**

This paper tackles the problem of object-centric learning in the video setting. The authors proposed to leverage existing video-slot attention architecture and add additional spatial and temporal attention blocks between current slots and previous slots to better model the spatial-temporal correspondence between frames. The authors leveraged a memory buffer for all history slot information considered. The resulting model achieves state-of-the-art results and shows consistent improvements over prior architectures on commonly used video object-centric learning datasets (MOVi).

**Strengths:**

The authors proposed an intuitive model for better modeling the spatial-temporal consistency between object-centric representations during video object-centric learning. Compared with previous models which mainly used simple models like self-attention for linking information between frames, this design considers longer history and more complex attention between slots at different time steps. By only adding this module, we can observe consistent improvement over prior architecture and achieving new state-of-the-art.

**Weaknesses:**

[-] Despite the performance improvement of adding this STATM module, the quantitative results of the backbone models seem to be exhibiting a rather big gap on several datasets (e.g. SAVi and SAVi++ on MOVi-E compared with results reported from the SAVi++ paper [here](https://browse.arxiv.org/pdf/2206.07764.pdf)). This hinders the evaluation of this paper's contributions, the authors might want to clarify the experimental settings to make these results more convincing.

[-] Following the previous point and as mentioned by previous works, slot attention seems very susceptible to the random initialization provided. The authors should consider reporting the results of several trials with different random seeds for better result presentation.

**Questions:**

See the Weakness section.

---

> ### Author Response · Authors · 2023-11-11
>
> Thank you for your review comments.  I will provide answers to the questions you raised.
> 1. In our paper, we explain our modifications to SAVi’s batch_size (from 64 to 32) and also to SAVi++’s batch_size (from 64 to 32) and training_steps (from 500k to 100k ). All other hyperparameters remain consistent with official settings. During our subsequent work, we realized that people might have questions about the results for the MOVi-E dataset. Therefore, we additionally trained the STATM module on the MOVi-E dataset with batch_size = 64 and training_steps = 500k (consistent with the settings in the SAVi++ paper). The results are mIoU=57.0 and FG-ARI=89.2, which exceeds the results reported in the SAVi++ paper( mIoU=47.1 and FG-ARI=84.1) and github (<https://github.com/google-research/slot-attention-video>, FG-ARI=85.1). We hope this provides you with better reference value and we will include these results in the supplementary material.
> 2. We currently have not used multiple sets of random seeds for experiments. However, it requires some time to address. In the future, we will continue to work on predictions, conduct experiments with multiple random seeds, and release code and the latest experimental results.
>
> Thank you once again for your valuable feedback.

---

### Official Review · Reviewer_km72 · 2023-10-31

**Soundness:** 2 fair
**Presentation:** 3 good
**Contribution:** 2 fair
**Rating:** 3
**Confidence:** 4

**Summary:**

This paper presents STATM (Slot-based Time-Space Transformer with Memory buffer), a method that extends slot-based scene representation methods SAVi and SAVi++ by incorporating a “memory buffer” or history of previous scenes, instead of a single scene, to update slot representations for objects in dynamic videos. STATM improves over smaller versions of SAVi and SAVi++ on the kubric benchmarks in terms of segmentation performance. Longer memory buffers help with segmentation performance particularly for the hardest dataset (MOVi-E in Kubric), both during training and at inference time. Several parallels between the STATM model and human cognition are drawn.

**Strengths:**

The paper is tackling an interesting problem (decomposing dynamic scenes / videos into objects), using a technique inspired by human cognition (people have working memory, and use motion cues to decompose scenes into objects). There is a large community of researchers interested in slot-based models, and this paper provides a general mechanism for how to improve slot-based scene representations by incorporating a history of previous time-steps.

The paper is reasonably easy to follow. The method is explained with sufficient detail to support reimplementation, and the figures explaining the method are helpful. The experimental results are presented clearly, and ablations over the memory buffer are sensible and help provide insight for how useful the memory is, and in which contexts. The qualitative segmentations are nice to see as well.

**Weaknesses:**

Overall, while the paper is tackling an established problem with a method inspired by human cognition, there are several issues that would need to be addressed before it could be impactful for the broader community. These can be generally grouped into:
* Adding comparisons to prior work that uses multiple history steps to infer segmentation masks
* Removing false statements about the SAVi/SAVi++ results
* Expanding explanation / background on SAVi for unfamiliar readers
* Toning down / removing incorrect / unverified statements about human cognition

## Adding comparisons to prior work that uses multiple history steps ##
There are now several papers that use multiple history steps with slot-based models to infer better segmentation masks. Three prominent ones are
* Wu et al, ICLR 2023, SLOTFORMER: UNSUPERVISED VISUAL DYNAMICS SIMULATION WITH OBJECT-CENTRIC MODELS
* Zoran et al, ICCV 2021, PARTS: Unsupervised segmentation with slots, attention and independence maximization.
* Chen et al, ECCV 2022, Unsupervised Segmentation in Real-World Images via Spelke Object Inference.

These methods all use dynamic information from videos in order to do better scene segmentation, and are highly related to STATM. The first two also have open source implementations (within the SlotFormer codebase, I think) that could be run on the Kubric MOVi benchmarks.

## Removing false statements about the results ##
There are some major issues with the presentation of the results that need to be addressed during a rebuttal. The first concerns *why* the SAVi models were trained in the manner they were (for up to 5x *fewer* iterations, and half the batch size). I understand that memory requirements may have prevented a larger batch size from being possible, but the number of training iterations and / or learning rate should be changed to compensate. The paper does neither, and as a result, the SAVi / SAVi++ results are *significantly worse* than what was reported in the original paper.

The paper states: “Due to constraints of computing resources, our models were trained for 100k steps with a batch size of 32, which differs from the official implementation of SAVi (small, 100k steps, batch size of 64) and SAVi++ (500k steps, batch size of 64). Nevertheless, under equivalent conditions, our models consistently outperform the original counterparts: e.g., STATM-SAVi (small, 100k steps, batch size of 32) achieves comparable performance to the official SAVi (large, 500k steps, batch size of 64) on MOVi-E datasets, while STATM-SAVi++ (100k steps, batch size of 32) performs comparably to SAVi++ (500k steps, batch size of 64)”

“STATM-SAVi (small, 100k steps, batch size of 32) achieves comparable performance to the official SAVi (large, 500k steps, batch size of 64) on MOVi-E datasets”
* This statement is false. Looking at the SAVi paper, on MOVi-E, SAVi obtains an mIoU of 30.7, while STATM-SAVi on the same dataset obtains an mIoU of 9.0. This is a 21+ point difference, not at all comparable.

“while STATM-SAVi++ (100k steps, batch size of 32) performs comparably to SAVi++ (500k steps, batch size of 64)”
* This statement is also false. From SAVi++ paper, the mIoU for MOVi-E is 47.1 vs. the 27.9 reported for STATM-SAVi++ in this paper, an almost 20 point difference.

These are dangerously misleading statements. Furthermore, the fact that the “small” versions of the model are performing dramatically worse than the larger models calls into question whether the “small” versions of the SAVi model are just behaving completely differently from the larger ones. If more training completely erases the advantage of STATM, what does that mean for the scientific conclusions? How should the broader community interpret this?

Given that it should be possible to train the model for longer, even if at a smaller batch size, I do not believe that the STATM results will meaningfully inform the community unless the base SAVi models are trained for longer, better approaching the original performance.

“On the MOVi-E dataset, the segmentation capability of the AS structure is not as robust as that of the CS structure, but it still outperforms the baseline”
* This is not true, at least based on comparisons to Table 1. The AS model is only doing better for FG-ARI, but is doing worse for mIoU.

## Expanding explanation of SAVi/SAVi++ ##

SAVi / SAVi++ are not explained enough for a reader not already familiar with the methods. The only mention of how these methods work is that there is a prediction and correction step, and that the correction step uses inputs to update the slots, and that STATM is replacing the prediction step. It would be helpful to further explain what the corrector module is doing, since the STATM method relies on this. However, the only information in the paper about this comes from the following two quotes:
* “The correction step uses inputs to update the slots”
* “... represents the slot information extracted from the corrector module of SAVi and SAVi++ at time t.”

This is not sufficient for a reader unfamiliar with SAVi to understand how the corrector module works, and what aspects of it are important for the STATM model.

## Toning down / removing incorrect / unverified statements about human cognition ##
The authors allude to human cognition throughout the paper as having inspired their architectural decisions for STATM. However, several statements made are either incorrect or unverified, and are therefore not helpful for bridging the cognitive science and machine learning communities. I list some examples below

“However, much like humans cannot predict the appearance of new objects in the next moment, the predictor faces similar limitations. At the initial moment, if the corrector cannot provide sufficiently accurate object information to the predictor, the predictor cannot offer precise prediction information for the corrector either.”
* These are not analogous situations, nor does it have anything to do with humans. No algorithm will be able to invent a correct appearance for an object if they do not have the sensor information for it.

In the section “Ablation Experiment of Memory Module.”:
“This aligns with human learning habits. Gathering more information at once is more conducive to humans in recognizing and summarizing patterns.”
* This is not necessarily true. Famous examples include “one-shot learning” from Lake et al, 2015.

“The model’s tracking and segmentation capabilities over a duration equal to the training frames are less affected by memory (see Figure 4b). This is analogous to a scenario where a person has observed a significant amount of object motion in various scenarios over a time duration t. Subsequently, when asked to predict or describe how objects move within that t time duration, as long as the inquiry doesn’t extend beyond t, the person should still be able to provide reasonably accurate predictions and explanations, even if their view is obstructed or their memory is restricted.”
* I did not understand this argument. Many papers have shown that people can extrapolate physics beyond “trained” time-frames trivially. Some of the cited works in the paper demonstrate that (Smith et al, 2019, Ullman et al, 2017) for example. If this claim is important, it needs to be backed up with a cognitive science citation.

**Questions:**

* How does this model compare to alternatives that also use information in time to do segmentation (the baselines presented in “weaknesses”)?
* What happens to the advantage of STATM if the base SAVi / SAVi++ models are trained for longer, to compensate for the smaller batch size?
* What are the critical connections to human cognition that you would like to make, and what are some citations for those?

---

> ### Author Response · Authors · 2023-11-11
>
> Thanks for your review comments. I will provide answers to the questions you raised.
> 1. Regarding the Q1, our initial intention was to create a model that emulates human-like behavior, capable of learning universal physical knowledge and utilizing it for predictions. However, during the prediction process, we realized that human reasoning plays a significant role in the perception of complex scenes. For instance, if there is a cat hidden in dense foliage, it might be challenging for us to detect it. Yet, if someone points out the cat's location, we can identify it. However, considering that the cat might be in motion, jumping onto a tree or into a thicket, this poses challenges for continued tracking and identification. Yet, if an individual can effectively understand the cat's movements (temporal reasoning) and anticipate its potential hiding places (spatial interaction), they can more easily continue identifying and tracking the cat. We drew inspiration from human "cat tracking" behavior, emphasizing the importance of prediction and reasoning in perception. Our core research focuses on human-like intuitive physics behavior, aligning well with SAVi and SAVi++. Since SAVi++ demonstrates SOTA results, our improvements are based on these models. While cited research aids our work, it slightly differs from our objectives and wasn't included in comparisons. We may consider comparisons on open-source platforms after completing prediction work. Thank you for your valuable insights.
> 2. Regarding the Q2, your comment on “STATM-SAVi (small, 100k steps, batch size of 32) achieves comparable performance to the official SAVi (large, 500k steps, batch size of 64) on MOVi-E datasets”, our expression may not be entirely accurate. We will make adjustments. What we intended to convey is that STATM-SAVi-small (with 960k+ parameters) achieves a comparable FG-ARI to SAVi-large (with 22340k+ parameters). Even under this unfair comparison, having a metric that is comparable already indicates an advantage of our model.
> Regarding the Q2, your comment on "from SAVi++ paper, the mIoU for MOVi-E is 47.1 vs. the 27.9 reported for STATM-SAVi++, an almost 20 point difference." We realized that people might have questions about the results for the MOVi-E dataset. Therefore, we additionally trained the STATM-SAVi++ module on the MOVi-E dataset with batch_size = 64 and training_steps = 500k (consistent with the settings in the SAVi++ paper). The results are mIoU=57.0 and FG-ARI=89.2, which exceeds the results reported in the SAVi++ paper( mIoU=47.1 and FG-ARI=84.1) and github (FG-ARI=85.1). This indicates that with longer training and larger batch size, the model's gains do not diminish. It is also evident that the model's results are much higher than those reported in the SAVi++ paper. We will include these results in the supplementary material.
> 3. Regarding the Q3, we briefly addressed it in the response to Q1. Our primary inspiration comes from literature on intuitive physics (Luis S Piloto, Ari Weinstein, Peter Battaglia, and Matthew Botvinick. Intuitive physics learning in a deep-learning model inspired by developmental psychology. Nature human behaviour), exploring the relationship between artificial intelligence and intuitive physics. Unfortunately, the paper did not provide open-source code, leading us to build our own model for learning intuitive physics. SAVi and SAVi++ are the closest existing models to our goals, hence our choice to improve upon them. Additionally, the lack of a detailed comparison in the cited literature is because it's not the primary focus of our current research. Our main goal is to validate the ideas presented in response to Q1.
> Regarding the statement "However, much like humans cannot predict the appearance of new objects in the next moment", this aligns with common knowledge. Humans indeed cannot predict if a completely new object will suddenly appear in their field of view in the next moment, similar to the unexpected entrance of a motorcycle into one's field of vision. People might react with surprise in such situations, which is an objective fact and a crucial aspect of intuitive physics. I'm having difficulty understanding the question you're trying to convey with the statement, "No algorithm will be able to invent a correct appearance for an object if they do not have the sensor information for it." If you could provide more context or clarification, I would be happy to assist further.
> 4. Regarding what you mentioned in the weakness, "On the MOVi-E dataset, the AS structure is outperforms the baseline". There is a problem with the expression. We actually want to explain the FG-ARI high , mIoU is almost the same. And AS is not the focus of this study.
> 5. Regarding the weakness of "Expanding explanation of SAVi/SAVi++", we will discuss it and make modifications accordingly.
>
> Thank you once again for your valuable feedback. If you have further questions, we'd be happy to discuss them with you.

---

> > ### Comment · Reviewer_km72 · 2023-11-23
> > **Thanks for the clarification, maintaining my score**
> >
> > Thank you for your replies.
> >
> > I would encourage the authors to take a look at the wording in the SAVI / SAVI++ papers. For every claim that is made in SAVI / SAVI++ about being inspired by human cognition, a citation is given. If the focus of the paper is to describe an intuitive physics system similar to what humans have, then citations to human cognition are needed. Not including these is misleading and unscientific. Specifically, as the authors note in the rebuttal "Our core research focuses on human-like intuitive physics behavior, aligning well with SAVi and SAVi++." This was not SAVI / SAVI++'s goal, their goal was to support object discovery. That is a very different goal from human-like intuitive physics.
> >
> > The combination of this issue, and the admittance "our expression may not be entirely accurate. We will make adjustments", in relation to statements about the model's comparison with SAVi / SAVi++, additionally without these adjustments being made, means that I am maintaining my score to reject.

---

> > > ### Author Response · Authors · 2023-11-23
> > > **Thanks for your reply**
> > >
> > > We would like to thank you for your picky reply. While part of your comments and suggestions are valuable, we hold completely different opinions on other picky ones. Anyway, thanks for your time and efforts placed on reviewing our paper. We will withdraw the paper and revise it for submission to another avenue.